# Phenological Features of the Spongy Moth, *Lymantria dispar* (L.) (Lepidoptera: Erebidae), in the Northernmost Portions of Its Eurasian Range

**DOI:** 10.3390/insects14030276

**Published:** 2023-03-09

**Authors:** Vasiliy I. Ponomarev, Georgiy I. Klobukov, Viktoria V. Napalkova, Yuriy B. Akhanaev, Sergey V. Pavlushin, Maria E. Yakimova, Anna O. Subbotina, Sandrine Picq, Michel Cusson, Vyacheslav V. Martemyanov

**Affiliations:** 1Institute Botanic Garden UB RAS, 8 Marta Str., 202a, 620144 Ekaterinburg, Russia; 2Institute of Animal Systematics and Ecology SB RAS, Frunze Str. 11, 630091 Novosibirsk, Russia; 3Department of Natural Sciences, Novosibirsk State University, Pirogova 2, 630090 Novosibirsk, Russia; 4Laurentian Forestry Centre, Natural Resources Canada, Quebec City, QC G1V 4C7, Canada; 5Biological Institute, National Research Tomsk State University, Lenina Str. 36, 63450 Tomsk, Russia

**Keywords:** pheromone monitoring, northern limit of the range, sum of effective temperatures, invasion, gypsy moth, climate change

## Abstract

**Simple Summary:**

Globalization accelerates the mixing of populations, including those of humans, animals and plants. Consequently, some species invade new natural communities and alter them. One illustrative example is the spongy moth, which was accidentally introduced into North America, where it became a major forest pest after its establishment there. In its native Eurasian environment, the spongy moth’s range has recently shown significant northward expansion, presumably as a consequence of global warming. In its North American range, a similar northward advance is yet to be fully documented. Eurasian populations of the spongy moth show significant developmental plasticity, which allows them to establish themselves in the much colder northern regions due to an acceleration of larval development. The mechanism responsible for this acceleration is not well understood, but our earlier studies point to genetic and epigenetic factors as playing a role. This plasticity indicates that new introductions of Eurasian spongy moths from the northernmost populations pose a more significant threat to North America than previously expected.

**Abstract:**

The spongy moth, *Lymatria dispar*, is a classic example of an invasive pest accidentally introduced from Europe to North America, where it has become one of the most serious forest defoliators, as in its native range. The present study was aimed at (i) identifying the current northern limit of *L. dispar*’s Eurasian range and exploring its northward expansion in Canada using pheromone trap data, and (ii) comparing northern Eurasian populations with those from central and southern regions with respect to male flight phenology, the sums of effective temperatures (SETs) above the 7 °C threshold necessary for development to the adult stage, and heat availability. We show that the range of *L. dispar* in Eurasia now reaches the 61st parallel, and comparisons with historical data identify the average speed of spread as 50 km/year. We also document the northern progression of *L. dispar* in southern Canada, where the actual northern boundary of its range remains to be identified. We show that the median date of male flight does not vary greatly between northern and southern regions of the spongy moth range in Eurasia despite climate differences. Synchronization of flight at different latitudes of the range is associated with an acceleration of larval development in northern Eurasian populations. Similar changes in developmental rate along a latitudinal gradient have not been documented for North American populations. Thus, we argue that this feature of spongy moths from northern Eurasia poses a significant invasive threat to North America in terms of enhanced risks for rapid northward range expansion.

## 1. Introduction

*Lymantria dispar* (Lepidoptera: Erebidae) is an important forest defoliator that can feed on more than 300 species of woody plants [1]. Populations of this irruptive species periodically reach outbreak proportions over vast areas [2,3,4,5], owing in part to this insect’s high ecological flexibility [6]. As a result, *L. dispar* is prominent on the list of the 100 most serious invasive species in the world [7]. The story of European spongy moths accidentally escaping from the laboratory of Leopold Trouvelot, in Massachusetts, followed by their permanent establishment in North America, has become a classic example of a successful invasion of a species from one continent to another [8]. 

The native range of the spongy moth is considered to extend latitudinally from ~20° N to ~58° N [9]. Although this univoltine species can develop under a wide variety of climatic conditions, individuals must undergo a winter diapause to complete their life cycle, and they do so as fully formed, pre-hatching 1st instar larvae. Thus, eggs are laid towards the end of the summer, but hatch only in the following spring, when tree leaves start expanding and are ready for consumption by the young larvae.

Most authors have reported April–May as the period during which spongy moth eggs hatch, irrespective of the study location’s latitude. Examples include Algeria (36° N), where eggs hatch in April [10]; Iran (36° N), in the last third of May [11]; the south of Kyrgyzstan (40° N), in the second third of April [12]; Hungary (47° N), in April [13]; Khabarovsk krai (48° N; Russian Far East), also from the end of April to the beginning of May [14]; and Kharkiv, Ukraine (50° N) and European Russia, from the end of April to the beginning of May [15,16]. In the northern portion of the spongy moth’s North American range, eggs hatch in mid-May in New Brunswick (46° N) and between the end of April and the middle of May in British Columbia (48° N), where populations are not yet permanently established [17]. Further south, egg hatch occurs in April in Virginia (37° N) [18]. According to a spongy moth life-stage model [19], egg hatch in US southern states could occur as early as March. Based on the aforementioned field records, however, we can say that spongy moth egg hatch typically takes place within a ~60-day time window spanning April and May, at least in the southern and central portions of the species’ range.

An examination of pheromone trap (males) and visual observation records shows that the *L. dispar* flight period displays variation similar to that described above for egg hatch in the southern and central latitudes of the range. For example, there is a tendency for flight to occur throughout June and July in some locations (Algeria: [10]; south of Kyrgyzstan: [12]), while it occurs later (end of July to mid-August) in other locations, such as Iran [11] and Ukraine [15]. Within Hungary, a country characterized by a complex relief, the spongy moth flight period varies as a function of the sampling location and year of observation, extending from June–July until the beginning of September [13], whereas in both European Russia and the Russian Far East, the flight period occurs during the months of July and August [14,16]. The situation is similar in North America, where flight occurs in June–July in the southern locations of the spongy moth range [18], whereas the period is shifted towards August and early September at higher latitudes [20,21]. In summary, adult flight tends to occur from June to the end of August or early September, depending on the latitude. 

The above data show a general trend for a shift in egg hatch and flight phenology towards later dates with increasing latitude. Spongy moths from Iran, however, seem to be an exception to this pattern, perhaps because of the significant divergence reported for Middle Eastern and Caucasian spongy moths relative to other *L. dispar* populations [22,23]. Thus, notwithstanding the above observation about the Middle Eastern populations, one might predict a significant shift towards later phenology for *L. dispar* development in regions near the northern boundary of its Eurasian range. In spite of all the research attention this insect has received, there are very limited data about its phenology in the northernmost parts of its range. The most relevant observations were made near Ekaterinburg (56°50′ N) [24], where egg hatch usually occurs in early to mid-May and male flight takes place in July–August; similar observations were made in Udmurtia (56°49′ N) [25]. These observations point to a very limited phenological shift (only one to two weeks), relative to dates recorded at the center of *L. dispar*’s range, despite a very significant latitudinal shift northward (i.e., >10°).

Owing to global climate change, heat availability in northern regions is likely to increase, which is predicted to positively affect the northward expansion of *L. dispar*’s range [26]. This prediction is supported by recent records documenting a northward shift of 200 km in the Asian spongy moth’s range [27]. Thus, spongy moth phenological data in the northernmost portions of its range are essential to assessing the possibility of outbreak spread following an expansion of its range. We do not consider photoperiodic conditions as playing an important role in the phenology of spongy moths because this species has an obligate diapause, and its development does not depend on daylength [28].

The aim of this study was to compare the phenology of male flight in parts of the spongy moth’s Eurasian and North American ranges. As part of this aim, (i) we assessed the current position of the northern boundary of *L. dispar*’s Eurasian range and examined its northward expansion in North America, using pheromone trap data; (ii) we compared the male flight phenology of populations in northern Eurasia with populations from central and southern regions of Eurasia and sums of effective temperatures (SETs) above the 7 °C threshold required for development to the adult stage in all three regions with corresponding data on heat availability. 

## 2. Materials and Methods

To monitor male flight, we used milk carton traps, each equipped with a pheromone dispenser loaded with 500 µg of (+)-disparlure (Scentry Biological, Inc., Billings, MT, USA), and an insecticidal strip containing 2.2-dichlorovinyl dimethyl phosphate (Hercon Environmental, Emigsville, PA, USA). We used slightly different methods for monitoring the presence of male moths, depending on the research question being addressed (i.e., estimating of northern boundary vs. phenology of flight).

To estimate the current position of the northern boundary of the spongy moth’s range in Asia, pheromone traps were secured onto host trees at a height of 1.5–2 m at the end of June and removed at the beginning of August in 2020 and 2021. We used one trap per sampling location. Traps were not left in the field for the entire flight season to avoid DNA degradation of captured moths, which we intended to use for subsequent population genomics analyses (in the context of a distinct study). Thus, the total number of adults caught in a given trap should not be taken as an accurate indicator of moth density in that study area (Figure 1); however, it provides qualitative data about the presence of moths in that area and still shows trends in the status of populations. Indeed, at the time traps were taken down from the trees, male flight had already reached its declining phase, suggesting that our trapping activity had covered most of the flight period. It is unlikely that the presence of female moths significantly affected captures of males by pheromone traps due to competition between females and pheromone lures when population levels are high; the latter is suggested by the fact that pheromone traps caught very high numbers of males in an independent study of spongy moth flight phenology [29]. In Canada, the traps were secured to host trees in a manner similar to that described above. Monitoring was conducted in 2017 and 2021, from the end of June to the end of September, a period that encompasses the entire flight season. We used two traps per location, each separated by 15 m. The use of a second trap was meant to prevent loss of monitoring data for a given site in the event of accidental damage to a trap. The total number of moths per site was calculated at the end of the flight season. For detailed trap locations, see Figure 1 and Figure 2. 

To assess seasonal male flight dynamics, pheromone-based monitoring in the northern portion of the spongy moth’s Eurasian range was carried out in the following localities: Ekaterinburg and surroundings (Sverdlovsk oblast, 56°51′ N, 60°36′ E) from 2010 to 2021 (last outbreak: 2005–2011); Tobolsk (Tyumen oblast, 58°11′ N, 68°15′ E) in 2020 and 2021 (last outbreak: 2016–2018); Kyshtovka (Novosibirsk oblast, 56°33′ N, 76°37′ E) from 2019 to 2022 (last outbreak: 2018–2020). For comparative purposes, monitoring was also conducted in the central and southern portions of the range, using the same protocol. The central locations were: Karasuk (Novosibirsk oblast, 53°74′ N, 78°00′ E) in 2013 and 2022; Astrakhan oblast (46°20′ N, 48°02′ E) in 2011, 2013 and 2014 in Volgograd, Volgograd oblast (48°43′ N, 44°30′ E), from 2010 to 2012, and in 2014. The southern locations were: Bishkek, Kyrgyzstan (42°52′ N, 74°35′ E) in 2009, 2012 and 2013; Jalal-Abad, Kyrgyzstan (40°56′ N, 73°00′ E) in 2004, 2005 and 2017; Alma-Ata, Kazakhstan (43°15′ N, 76°55′ E) in 2007 (for the positions of these locations on a map, see Figure 3). With the exception of Tobolsk (Tyumen oblast), Kyshtovka (Novosibirsk oblast) and Jalal-Abad (Kyrgyzstan, in 2004 and 2005), where only one trap was deployed per location, 3 to 10 traps were used to monitor seasonal flight dynamics at each location. At any given location, the distance between traps varied between 200 m and 2 km. In all cases, the traps were hung to host trees in mid-June (before the start of male flight) and taken down at the end of September. Captured males were counted at a minimum frequency of every four days.

To compare male flight dynamics among different trapping locations, we used the median date of male flight, which is taken here to coincide with the emergence of ~50% of adults at a given location.

The attraction range of the *L. dispar* pheromone is considered to vary between 300 m and 15 km [30]. However, it has been suggested that atmospheric transport of male moths could increase the apparent attraction range of the pheromone beyond 15 km [31]. There has been only one report of a long-range spongy moth flight (100 km), but substantiating evidence for this claim is lacking [32]. All of the above considered, the geographic scale of our study (Figure 3) makes movement of male moths between sample locations highly unlikely.

**Figure 3 insects-14-00276-f003:**
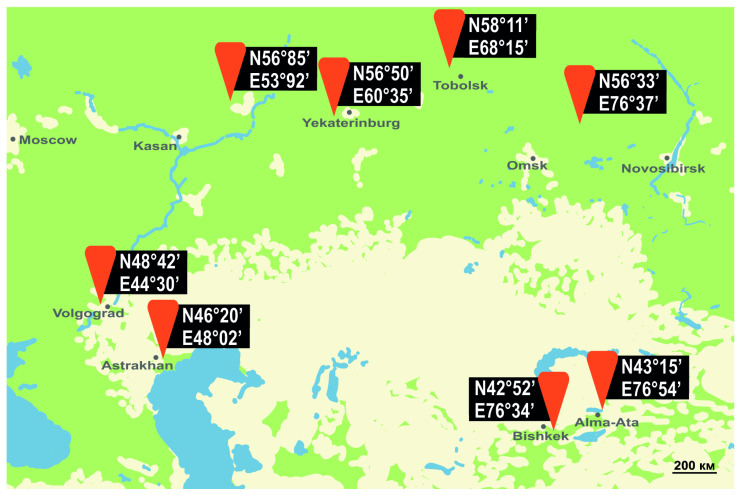
Map of localities where flight data were recorded for calculating the median date of *L. dispar* male flight.

Taking into account the fact that the curve of the dynamics of male flight is nonparametric, which is confirmed by calculations carried out by us using the Kolmogorov–Smirnov criterion to characterize the flight of imago and calculate the SET of their development, the median date of male flight was taken to represent the departure of about 50% of adults in a given locality. 

We used the median date of male flight as a checkpoint to calculate the sum of effective temperatures (SET) needed for development to the adult stage. Since insects are poikilothermic animals and their development is affected by temperature, a lower development threshold (LDT) is required to calculate both the SET for individual development and the heat availability of the growing season in the region where monitoring is being carried out. Different LDTs have been reported in the literature for the spongy moth, varying as a function of life stage and geographic population. While different thresholds for pre-hatching spring development of embryos have been reported (3 °C to 7 °C; [15,33,34]), larval development thresholds are considered to vary between 6 °C and 10 °C [15,34,35,36,37]. Pupal LDT, on the other hand, has been estimated to be 10 °C [38], whereas that of early embryo development in late summer has been assessed to be 6.8 °C [38] or 7 °C [16,34]. Estimation of the latter threshold for the North American “NJSS” laboratory line yielded a slightly higher value (10 °C), based on the duration of early embryonic development measured at 15 °C and 25 °C [39]. It can be seen from the foregoing that the choice of a unified criterion for validly comparing the phenology of different stages of development and different populations is not a straightforward task. For this reason, we opted to rely on the work of Kozhanchikov [38] and Ilyinsky et al. [16], who assessed average SET values for Eurasian spongy moth populations (i.e., from the present study area) using an LDT of 7 °C. They estimated the SET for full development from late embryos (in spring) to imago (in mid-summer) to be 990 and 930 degree-days for females and males, respectively, while the summer–autumn development of embryos required 300–310 degree-days, resulting in a SET of 1230–1300 degree-days for the entire life cycle. 

For calculation of the SET, we used the 25 °C value whenever average daily temperatures were >25 °C but <32 °C, as temperatures > 25 °C have been shown to have no significant effect on the developmental rate of the spongy moth [37,38]. However, temperatures exceeding the upper developmental threshold for this species (32 °C; point at which thermal dormancy is induced) were excluded from the SET calculation [36,38]. Such corrections were applied to only a limited number of southern monitoring locations, namely, Bishkek, Kyrgyzstan (42°52′ N, 74°35′ E), and Alma-Ata, Kazakhstan (43°15′ N, 76°55′ E), where (as shown in the present work) the SET is significantly higher than in other parts of the species’ range. This procedure avoided overestimation of the SET in these monitoring locations, while in locations further north, the average daily temperature never exceeded 25 °C. We calculated the early-embryonic SET using the median date of male flight as starting date and the date marking sustained transition below the temperature threshold of development as end date. This was done for each trapping locality to estimate the potential conditions available for early embryo development before overwintering. In addition, our previous work revealed a drop in late embryonic and larval developmental rates in response to an increase in early-embryonic SET [24]. 

To assess the suitability of weather conditions for the maintenance of a population in a given study area, we calculated the heat availability of the growing season (also with a 7 °C threshold), starting in the spring, following the stable transition of temperatures to a point above the threshold for at least a week, and ending in the autumn, when temperatures fell below the threshold. This was done for each trapping locality to estimate the potential conditions available for early embryo development before overwintering. Average long-term heat availability was calculated for the last 11 years from 2010 to 2021. We calculated the SET and heat availability using temperature data from weather stations for Eurasian localities [40,41]. For North American localities, we only assessed heat availability using data from the Government of Canada’s National Weather Service [40,42,43]. 

## 3. Results

### 3.1. Northern Limit of L. dispar in Eurasia and North America

We estimated the position of the northern boundary of the spongy moth’s Asian range by monitoring adult males, using pheromone traps placed along south-to-north transects in Siberia. Trap catches recorded in 2021 (Figure 1) indicate that the northern edge of the range lies between 61° N and 62° N, which is considerably further north than the boundary reported earlier for the same longitude (see Figure 4, produced using data from Ilyinykh and Krivets [27]). This difference represents a northward expansion of 5° in 10 years. Comparisons with other early estimations of the northern boundary of this species’ range likewise point to a significant northward expansion [9,38]. Thus, earlier predictions regarding the northward expansion of the spongy moth’s range in Eurasia [26] appear to be verified by the present work. Although contours of this species’ range are known to be unstable and could pulse over long periods of observations [21], the use of several transects in our study and the comparability of our data with those of Ilyinykh and Krivets (similar trapping methods and locations) indicate that the northward expansion of *L. dispar*’s range is truly an ongoing process, in parallel with climate change [20]. Whereas the spongy moth’s northward expansion front seems to lie between the 61st and 62nd parallels (see above), the latitude beyond which populations can no longer realize their full biological potential and develop outbreaks lies further south, between the 57th and 58th parallels [5]. 

Pheromone-based monitoring in North America did not allow us to reach the northern edge of the spongy moth’s range in eastern Canada; indeed, traps as far north as 48.7° were still catching moths (Figure 2). However, when considering data from the two sampling years presented here (Figure 2) and range maps published earlier [44], it is clear that the spongy moth is undergoing a northward range expansion in southern Canada. Importantly, one of the northernmost traps (#11) caught a very significant number of moths (100 moths per season; Figure 2), suggesting that the northern boundary of the North American range could be significantly further north. However, it is not yet clear whether the moths we caught there were from locally established populations or developed from eggs accidentally transported to this location in 2021.

### 3.2. Flight Phenology, SET and Heat Availability

Pheromone-based monitoring data published earlier for Ekaterinburg, covering the period 2009–2015 [24], are here updated with data for the period 2016–2019 (Table 1). The male flight median for Trans-Ural (eastern foothills of the Ural) populations (Ekaterinburg city, Sverdlovsk oblast) generally falls in the last two-thirds of July. Other northern populations, such as those of the Cis-Ural (western foothills of the Ural; Udmurt Republic) and West-Siberia (Tobolsk, Tyumen oblast and Kyshtovka, Novosibirsk oblast), display a male flight median within the same period. Thus, examination of male flight data across latitudes that span ~42° N to ~58° N in the northern Eurasian range (Table 1) shows that the median flight date typically falls within the second half of July, irrespective of latitude, despite a later egg hatch in the northern regions (up to a 60-day difference relative to southern regions). Although statistical analysis indicated that male flight occurred significantly later in northern than in central and southern localities (Kruskal–Wallis H-test: H (2, 109) = 60.31 *p* < 0.0001; Figure 5), presumably as a result of later spring warming in the north, the median dates of male flight remained in the range of ~7 to 21 July.

Typically, males emerge earlier in the season than females [1]. While pupation dates of males tend to be 4 to 5 days earlier than those of females, the latter go through the pupal stage 1 to 3 days faster than males. As a result, the difference in emergence time between males and females is about 2 days, in both laboratory cultures and wild populations [45]. According to Thompson et al. [46], the differences in the duration of the development period of males and females till the adult stage in North American populations range from 5 to 13 days, depending on the rearing temperature and origin. According to Limbu et al. [37], for Eurasian populations that have reached 50% of adult emergence, the difference in SET between males and females is about 40 degree-days, which at an 8–9 °C threshold and a 20–25 °C rearing temperature, takes 2 to 4 days. Thus, male flight dates can be used to estimate the date of completion of the adult stage in a given region and to assess the duration of embryo development prior to overwintering.

As is true of other arthropods, the development of the spongy moth is dependent on the amount of heat it receives during the summer season, and a minimum amount of heat accumulated during development is necessary for the successful completion of its life cycle. Thus, heat availability is an important parameter to assess the likelihood of spongy moth establishment in new regions located near the northern boundary of its range. In the vicinity of the northern edge of the spongy moth’s range, the long-term average heat availability during the growing season, based on a development threshold of 7 °C, varies between 1045 ± 47 degree-days (x ± se) in Khanty-Mansiysk (coordinates: 61°0′ N, 69°0′ E), 1042 ± 48 degree-days in Surgut (61°15′ N, 73°25′ E) and 1020 ± 48 degree-days in Nyzhnevartovsk (60°56′ N, 76°33′ E). However, equivalent values are higher (~1150 to 1350 degree-days) in northern areas where populations can reach outbreak proportions. More specifically, the average heat availability for Ekaterinburg was 1337 ± 89 degree-days, while it was 1132 ± 33 for Tobolsk and 1155 ± 45 for Kyshtovka (Table 2). According to the literature, early embryonic development in late summer requires at least 300 degree-days of SET for successful overwintering. Completion of full development in the next spring–summer period requires an additional ~930 and ~990 degree-days for males and females, respectively (at the lower development threshold of 7 °C). Based on these data, the full life cycle of the spongy moth requires about 1250–1300 degree-days for its completion [16,38]. Surprisingly, however, heat availability in the northern portions of the spongy moth’s range, where outbreaks can develop, is often below these values (Table 2). Nevertheless, populations in these regions seem stable and periodically form outbreaks.

Our data (Table 2) show a gradual south-to-north decline in the SET for Eurasian spongy moth populations. SET values reported here for the central portion of the Eurasian range (Volgograd) are comparable to those reported in the early literature [16,38]. In the southern regions (Alma-Ata and Bishkek), we see that the SET increases with the observed rise in heat availability, while the opposite is true for the northern regions. Thus, our data indicate that the developmental SET displays considerable plasticity in this species, varying as a function of regional heat availability. 

The SET required for development from late embryo to imago varied significantly among northern, central and southern populations (Kruskal–Wallis, H (2, 109) = 67.20; *p* < 0.0001; Figure 6), indicating that northern populations require less heat than populations located further south to maintain the same growth rate.

## 4. Discussion

The present study raises questions as to what mechanism is responsible for the significantly reduced developmental period, up to the imago stage, reported here for spongy moths found in the northernmost portion of the species’ Eurasian range, in comparison to values obtained for populations in the central and southern regions. In a previous study, we found an increase in the rate of larval development, especially in younger instars, associated with a reduction in the summer–autumn SET, i.e., during early embryonic development and the forepart of diapause, before the onset of low temperatures [24]. Thus, when we experimentally lowered the SET for early embryonic development and diapause before overwintering (430 versus 1230 degree-days) in insects collected in the northern (Ekaterinburg) and central (Volgograd) populations of *L. dispar*’s range, we observed a decrease in the duration of larval development for the northern group in both males (44.1 ± 0.8 days vs. 53.7 ± 3.5 days) and females (48.3 ± 1.2 vs. 54.4 ± 1.2), which was mainly due to a reduction in the duration of younger instars (1–3). For the central population, a decrease in the duration of larval development was observed only in females (61.8 ± 1.7 vs. 67.9 ± 2.3).

This could be an adaptive feature of the northern populations, which often face a heat deficiency with respect to the summer–autumn SET, relative to the value that is necessary for successful overwintering (300 degree-days, based on Ilyinsky [16]). Indeed, estimates of the SET provided in Table 2 show that, in northern populations, the summer–autumn SET (i.e., for the period from egg laying until the onset of cold weather) is often lower than 300 degree-days, even when calculated based on the flight median (see value for Ekaterinburg, year 2014; Table 2). In southern regions, however, weather conditions are not a limiting factor for early embryonic development. A lowering of the developmental SET in populations near the northern boundary of the *L. dispar* range could either be the result of selective pressures favoring individuals with faster development or favoring individuals with greater developmental plasticity. The latter hypothesis is supported by our earlier experimental data showing that the larval developmental period can be modified by artificially changing the conditions of embryo development in an environmental chamber [24]. 

The above observations suggest that differences in the developmental rate may have a genetic basis, irrespective of whether these characters are fixed or are the result of greater plasticity. Based on the existing literature (e.g., Limbu et al. [37]), the thresholds and SET for larval development of spongy moth can vary significantly among laboratory rearings when insects have gone through different numbers of generations. Not surprisingly, phenological models developed based on the data often produce estimates that do not agree with pheromone-based records obtained through monitoring in Asian ports [47], where male flight can occur later than what the model predicts. Such discrepancies support the existence of genetic-based differences in development between initial wild populations and laboratory lines.

In North America, the northern edge of the spongy moth range is considered to be located at ~48–49° N [17,21,44] (confirmed by observations in the present study), which is considerably further south than the northern boundary of the Eurasian range. According to information summarized by Régnière and Sharov [32], the median date of male flight in the southern United States (Virginia, West Virginia, North Carolina) tends to occur during the last three weeks of July, which fits well with predictions made by a phenological model based on SET calculations [48]. The long-term average heat availability calculated from data collected at the Chesterfield weather station (Virginia, 37°22′ N, 77°30′ W) was 2549 ± 38 degree-days (7 °C threshold), which is comparable with the heat availability in Astrakhan, Alma-Ata and Bishkek in Eurasia (2593 ± 35, 2535 ± 22 and 2669 ± 28 degree-days, respectively). In Bellingham, Washington (48°45′ N, 122°29′ W), closer to the northern boundary of the spongy moth’s range in North America, the observed median date of male flight (monitored in 1985–1986) fell during the last 3 weeks of August; this, again, generally agrees with the predictions of a phenological model. The average heat availability for that locality is 1462 ± 32 degree-days, which is considerably higher than at the sites where *L. dispar* pheromone monitoring was conducted in the northern regions of the Eurasian range, and where the median date of male flight usually occurs one month earlier, in the last third of July. It must be pointed out, however, that the later date of male flight in Bellingham, where the species is not considered to be established, is based on only two years of pheromone monitoring. Another study whose results are relevant to the issue at hand is one conducted in northern Minnesota [21], where male flight was monitored from 2005 to 2009, yielding a median date of male flight in the last 3 weeks of August, except for 2006, when it took place in the first third of August. Heat availability in Duluth, Minnesota (46°48′ N, 92°08′ W), was 1453 ± 42 degree-days, which, again, is higher than at the sites where we conducted our monitoring in the northern regions of the Eurasian range.

A comparison of our results (Figure 2) with previous observations in Canada [44] confirms the stable presence of *L. dispar* in the southern portions of Canada, from 1990 up to now. The northern boundary of the range runs from Charlo, New Brunswick (N 47°59′, W 66°19′), with an average heat availability of 1246 ± 71 degree-days, to Rouyn-Noranda, Quebec (48°25′ N; 79°03′ W), and Sault Ste. Marie, Ontario (46°31′ N, 84°21′ W), where the average heat availability is 1291 ± 22 and 1348 ± 38 degree-days, respectively. However, the relatively high number of individuals that we caught in trap #2 (Figure 2) suggests that the northern boundary of the range in Quebec could well be a few hundred km further north. Either the same or lower levels of heat availability are not unusual for the northern regions where we conducted our monitoring in Siberia, where *L. dispar* populations periodically reach outbreak levels. Outbreaks in Canada have been recorded south of the 47th parallel [44], where heat availabilities are >1400 degree-days (e.g., Sudbury, Ontario, 46°37′ N, 80°48′ W: 1438 ± 33 degree-days).

Based on the literature pertaining to *L. dispar* male flight in North America, there does not seem to be a significant reduction in the SET for development near the northern edge of the range, although some differences in the mass and rate of development of caterpillars have been reported [46,49], pointing to ongoing adaptive processes occurring in populations of that region.

As mentioned above, the finding of the reduction in the developmental SET among Asian spongy moth populations near the northern boundary of their range might be a genetically determined feature of these populations. In further support of this hypothesis is a recent nuclear-marker-based population genomics analysis of *L. dispar* across its entire geographic range [23], which shows that Siberian populations (corresponding approximately to the northern populations considered here) form a very distinct cluster from North American populations (i.e., high pairwise FST values). Since a reduction in the developmental SET near the northern edge of the spongy moth’s range is observed in both the European (Udmurtia) and Asian (Sverdlovsk, Tyumen and Novosibirsk oblasts) portions of the Eurasian range, we suggest that northern Eurasian populations display greater developmental plasticity in response to environmental factors than their North American counterparts, a trait that should be considered an important element of risk for rapid invasion in North America, in addition to the well-known female flight capability trait [50]. Thus, in assessing the threat posed by potentially invasive populations of *L. dispar* in North America, it is important to consider not only female flight capability [51] and host range [6], but also the potential for individuals to adapt to conditions of low heat availability, which could affect the rate of spread in the newly invaded environment (see also study by Ilyinykh and Krivets [27]). The ability of northern Eurasian populations to feed on conifers is the focus of a separate study by our group [52], demonstrating the clear potential for range expansion towards the taiga. 

## 5. Conclusions

Pheromone-based monitoring of male flight near the northern boundary of *L. dispar*’s Eurasian range allowed us (i) to document the significant northward expansion of this species and (ii) to show that the median date of male flight in the northern portions of the range, although somewhat later than observed in regions located much further south, is not delayed to the extent one would expect from the significant reduction in the SET seen in northern populations. This partial synchronization of flight in different latitudinal portions of the range is associated with an acceleration of larval development of northern Eurasian populations. The reasons for this accelerated development are currently unknown, but both genetic and epigenetic factors should be considered.

Based on the available literature, there is no evidence of a similar plasticity in developmental rate along a latitudinal gradient among North American populations. Thus, we argue that this feature of spongy moths from northern Eurasia poses a significant invasive threat to North America in terms of enhanced risks for rapid northward range expansion.

## Figures and Tables

**Figure 1 insects-14-00276-f001:**
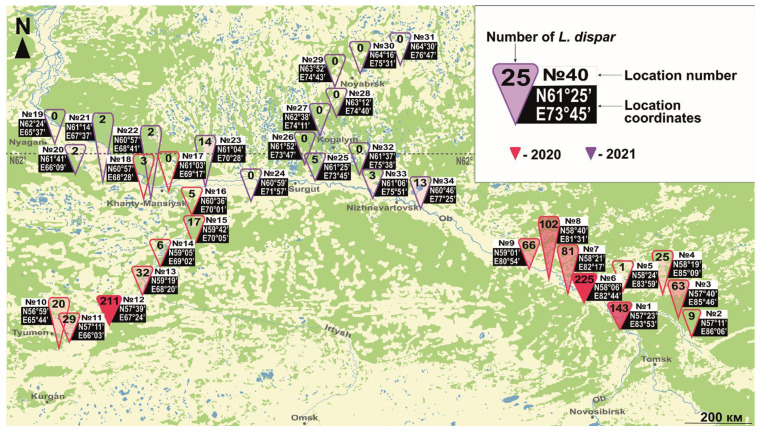
Results of pheromone monitoring of *L. dispar* in the northern portion of its range in Asia. Description of values is given in the upper right corner.

**Figure 2 insects-14-00276-f002:**
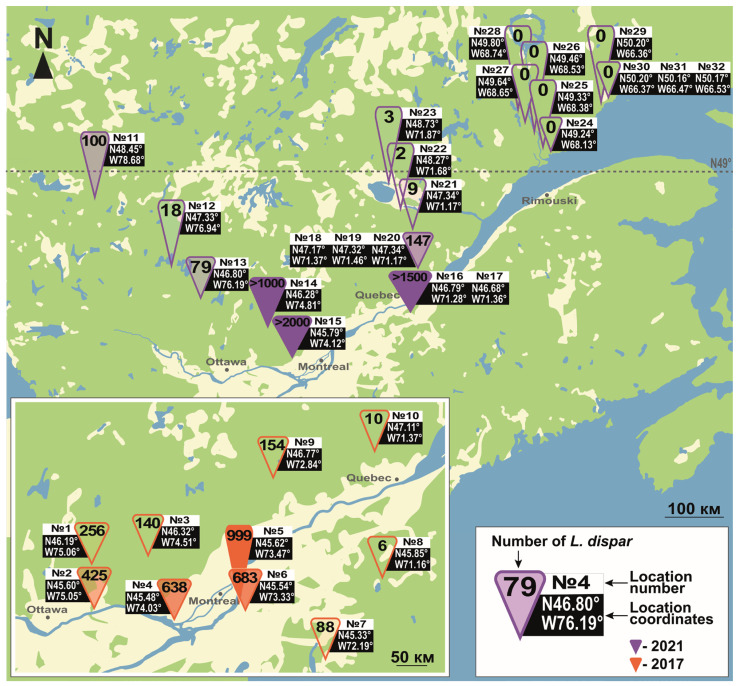
Results of pheromone monitoring of *L. dispar* in the northern portion of its range in North America. Description of values is given in the lower right corner.

**Figure 4 insects-14-00276-f004:**
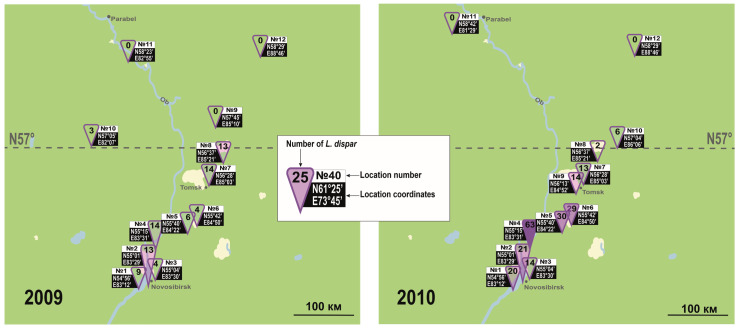
Results of pheromone monitoring of *L. dispar* in Western Siberia in 2009–2010, created using data from Ilyinykh and Krivets 2011 [27].

**Figure 5 insects-14-00276-f005:**
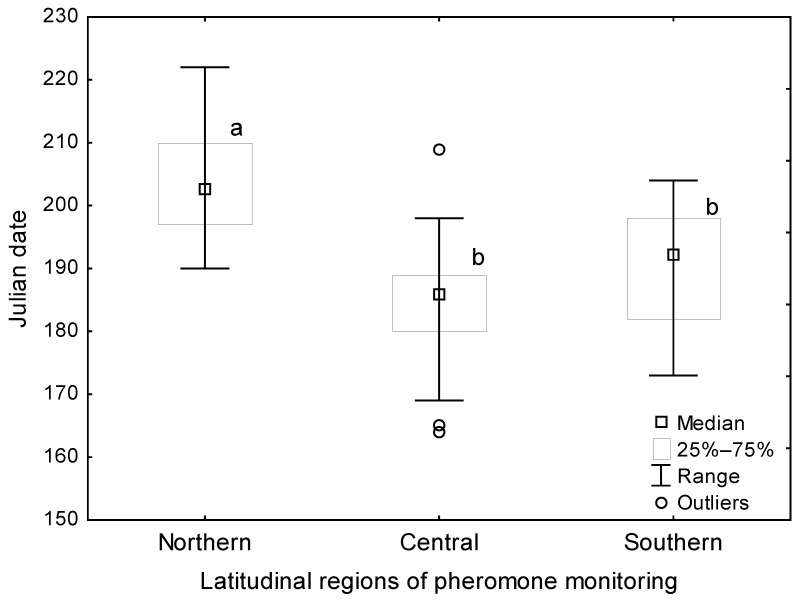
Median Julian date of *L. dispar* male flight for the localities flagged in Figure 3. Different letters indicate a significant difference between the localities being compared.

**Figure 6 insects-14-00276-f006:**
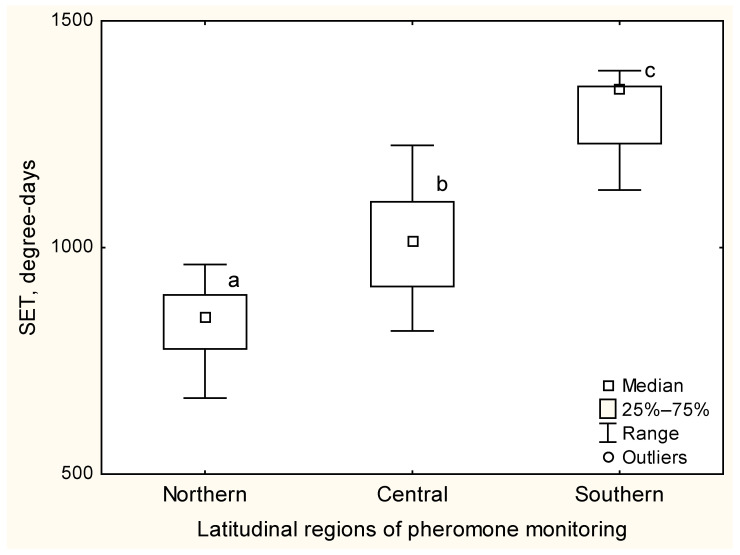
SET for development of *L. dispar* populations from the localities flagged in Figure 3, as calculated on the median date of male flight. Different letters indicate significant differences among the localities being compared.

**Table 1 insects-14-00276-t001:** Seasonal flight data for male spongy moths caught in pheromone traps at different latitudes in Eurasia.

Year	Flight Dates			Flight Duration, Days	Number of Males, Per Trap
	Start	End	Median		
58°11′ N–68°15′ E—Tyumen region, Tobolsk, Siberia					
2020	12 July	22 July	16 July	11	65
2021	12 July	12 August	20 July	32	49
56°85′ N–53°92′ E—Republic of Udmurtia, Votkinsky district, biostation “Siva”, Ural ^1^					
2014	15 July	9 August	2 August	26	20
2015	5 July	5 August	22 July	32	18
2016	7 July	1 August	12 July	26	12
56°50′ N–60°35′ E—Sverdlovsk region, Ekaterinburg, Ural ^2^					
2009	23 July	10 August	25 July	18	176
2010	3 July	7 August	22 July	36	182
2011	19 July	12 August	28 July	25	30
2012	3 July	18 July	11 July	16	17
2013	25 July	14 August	29 July	21	799
2014	31 July	19 August	9 August	20	7
2015	16 July	24 July	23 July	9	3
2016	8 July	31 July	20 July	24	575
2017	25 July	28 August	8 August	35	66
2018	1 August	6 August	2 August	6	4
2019	15 July	19 July	17 July	5	2
56°33′ N–76°37′ E—Novosibirsk region, Kyshtovka village, Siberia					
2019	- ^3^	12 August	30 July	~20	3399
2020	6 July	26 July	17 July	20	445
2021	5 July	8 August	15 July	35	1166
2022	11 July	14 August	31 July	33	4223
53°74′ N–78°00′ E—Novosibirsk region, Karasuk, Siberia					
2013	10 July	16 August	29 July	37	585
2022	7 July	4 August	12 July	28	1454
48°42′ N–44°30′ E—Volgograd region, Volgograd, Eastern Europe					
2010	2 July	31 July	6 July	30	14
2011	8 July	28 July	15 July	21	46
2012	7 July	27 July	18 July	21	152
2014	19 June	23 July	4 July	35	654
46°20′ N–48°02′ E—Astrakhan region, Astrakhan, Eastern Europe					
2011	28 June	5 August	9 July	39	814
2013	14 June	29 July	6 July	45	424
2014	18 June	20 July	1 July	33	478
43°15′ N–76°54′ E—Kazakhstan, Alma-Ata, Central Asia					
2007	3 July	3 August	15 July	31	170
42°52′ N–76°34′ E—Republic of Kyrgyzstan, Bishkek, Central Asia					
2009	3 July	8 August	23 July	37	409
2012	25 June	30 July	24 July	36	2956
2013	24 June	5 August	18 July	43	367
40°56′ N–E73°00′ E—Republic of Kyrgyzstan, Jalal-Abad, Central Asia					
2004	27 June	18 July	6 July	21	57
2005	21 June	24 July	6 July	33	164
2017	11 June	26 July	23 June	45	4498

^1^ Data from Ermolaev et al. [25]; ^2^ Pheromone monitoring was also conducted in Ekaterinburg in 2020 and 2021, but population levels were too low to be informative (only one individual caught in the last third of July); ^3^ 564 males caught on the first day trap was inspected.

**Table 2 insects-14-00276-t002:** Sum of effective temperatures (SET; degree-days) for the development of the spongy moth and seasonal heat availability at different latitudes in Eurasia (threshold: 7 °C).

Year	SET for Development to Imago Stage			SET		
	At Beginning of Flight	At End of Flight	At Flight Median	Summer–Autumn Embryo Development		Heat Availability
				At Flight Median	At End of Flight	
58°11′ N–68°15′ E—Tyumen region, Tobolsk (WMO id 28275 ^1^)						
2020	705	865	771	596	502	1367
2021	715	1017	776	517	276	1293
56°85′ N–53°92′ E—Republic of Udmurtia, biostation “Siva” (WMO id 28413)						
2014	756	1014	910	461	357	1371
2015	698	1004	850	551	397	1401
2016	686	1066	782	904	620	1686
56°50′ N–60°35′ E—Sverdlovsk region, Ekaterinburg (WMO id 28440)						
2009	738	892	757	549	414	1306
2010	659	1176	897	564	285	1461
2011	700	989	835	449	295	1284
2012	781	929	900	744	715	1644
2013	826	1067	880	491	304	1371
2014	763	976	868	273	165	1141
2015	713	786	776	411	401	1187
2016	687	989	848	769	628	1617
2017	655	1041	824	347	130	1171
2018	766	816	777	426	387	1203
2019	667	723	691	537	505	1228
56°33′ N–76°37′ E—Novosibirsk region, Kyshtovka village (WMO id 29405)						
2019	-	846	709	403	265	1111
2020	646	923	801	547	416	1348
2021	563	921	669	538	286	1207
2022	603	965	817	356	208	1173
53°74′ N–78°02′ E—Novosibirsk region, Karasuk (WMO id 29814)						
2013	554	1043	816	471	244	1287
2022	736	1211	909	710	408	1619
48°42′ N–44°30′ E—Volgograd region, Volgograd (WMO id 34560)						
2010	974	1374	1102	1296	931	2305
2011	936	1364	1112	1020	768	2132
2012	738	1029	887	1564	1422	2451
2014	787	1264	957	1247	940	2204
46°20′ N–48°02′ E—Astrakhan region, Astrakhan (WMO id 34880)						
2011	788	1533	1085	1243	795	2328
2013	792	1503	1191	1247	935	2438
2014	834	1429	1194	1318	1083	2512
43°15′ N–76°54′ E—Kazakhstan, Alma-Ata (WMO id 36870)						
2007	1074	1534	1203	1194	863	2397
42°52′ N–76°34′ E—Republic of Kyrgyzstan, Bishkek (WMO id 38353)						
2009	911	1533	1237	988	692	2225
2012	1097	1721	1342	1288	909	2630
2013	881	1604	1195	1222	813	2417
40°56′ N–E73°00′ E—Republic of Kyrgyzstan, Jalal-Abad (WMO id 38613)						
2004	1175	1544	1337	1672	1465	3009
2005	1080	1710	1350	1739	1379	3089
2017	898	1694	1114	1357	1937	3051

^1^ Identification number of meteostation by World Meteorological Organization.

## Data Availability

The data presented in this study are available on request from the corresponding author.

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
