# Peer review of "Phenological Features of the Spongy Moth, *Lymantria dispar* (L.) (Lepidoptera: Erebidae), in the Northernmost Portions of Its Eurasian Range"

_insects, 2023, doi:10.3390/insects14030276_

Round 1

Reviewer 1 Report

Acceleration of development in high latitudes may be a widespread and common phenomenon. It is certainly predictable as likely to evolve, given the relative unpredictability of onset of adverse conditions late in the season--one wishes to get into a secure overwintering phenophase as early as practicable as a matter of bed-hedging insurance. By focusing on latitudinally-expanding populations, this paper may have caught the process in medias res--good show! Loys of suggestions for future work.

Author Response

We kindly thank you for assessments of our manuscript and we are grateful that you found our work relevant.
We've tried to improve the language of our manuscript.

Kind regards

Authors

Reviewer 2 Report

The research topic of the authors is relevant. The scope of work is large. However, there are some comments on the presented material. Marked in yellow in the file.

lines 37–39. «Synchronization of flight at different latitudes of the range is associated with an acceleration of larval development in northern Eurasian populations»

– that's a question! dates of summer phenomena are very close at different latitudes within a territory with a pronounced change of seasons, where the timing and rate of development of plants and insects are determined by both temperature and photoperiodic response

lines 39–41. «Similar changes in developmental rate along a latitudinal gradient have not been documented for North American populations. Thus, we argue that this feature of spongy moths from northern Eurasia poses a significant invasive threat to North America in terms of enhanced risks for rapid northward range expansion».

– The range of the gypsy moth will still move north, provided there is sufficient heat and fodder plants.

lines 71–74. “...we can say that spongy moth egg hatch typically takes place within a ~60-day time window spanning April and May, at least in the southern and central portions of the species’ range”.

Within the entire range, the timing of larvae hatching is synchronized with the host plant foliage appearance. This occurs approximately after a stable temperature transition over 10 ° C. Within the stands, these dates may differ by several days, depending on the depth of soil freezing, the timing of its thawing, the beginning of the sap flow of trees, and the structure of stands. If the moths flew into another plantation, then in the next generations those caterpillars will survive, which hatching coincided with the leaves blooming in this particular stand.

Therefore, the sum of temperatures before the flight must be calculated not from the beginning of the year, but from the date of larvae hatching, approximately after the date of the stable temperature transition through 10 ° C. In each stand, the exact timing of larvae hatching can be determined by phenological indicators, for example, birch leaves blooming, Taráxacum flowering ...

Summer events (flight of L.dispar) generally vary little at different latitudes, to a greater extent within one stand.

lines 89–90. “The above data show a general trend for a shift in egg hatch and flight phenology towards later dates with increasing latitude”

– Spring in the north starts later, but the temperature rises faster than in the south, so by the middle of summer many phenomena are observed almost simultaneously at different latitudes. Closer to autumn, the photoperiodic reaction can contribute to the inhibition of the development of individual instars or stages.

lines 106–108. “Thus, spongy moth phenological data in the northernmost portions of its range are essential to assessing the risks of outbreak spread following an expansion of its range”.

Expansion of the range does not mean the ability to outbreak.

lines 109–110. Purpose ... goal ... aim?

– there are 3 tasks here: to find the northern border, to compare the flight dates, and to compare the SETs, and in the abstract there are 2 tasks (lines 28–32), the last two tasks are combined.

lines 115–116. “...above the 7°C threshold required for development to the adult stage in all three regions with corresponding data on heat availability.

Using the 7°C threshold to analyze summer events is counterintuitive. It is better to use a threshold of 10°C (see above) to determine the limit of species ranges. For example, this methodological approach was applied in predicting the distribution of Emerald ash borer to Northern Europe.

Orlova-Bienkowskaja, M.J.; Bieńkowski, A.O. Low heat availability could limit the potential spread of the Emerald ash borer to Northern Europe (Prognosis Based on Growing Degree Days per Year). Insects 2022a, 13(1), 52.

Webb, C.R.; Mona, T.; Gilligan, C.A. Predicting the potential for spread of emerald ash borer (Agrilus planipennis) in Great Britain: What can we learn from other affected areas? Plants People Planet 2021, 3, 402–413.

lines 131–132. The presence of males in a trap does not mean that there may be an outbreak in this stand. On the contrary, in high population density, male chooses female visually and prefers the natural pheromone to the synthetic one.

To verify the fact of L.dispar presence in the territory, the available data are sufficient. Trends in population dynamics can be traced from data obtained over several years, which cover all phases of the outbreak. Therefore, 2 years of records in the North American continent with a gap are sufficient only to conclude about the penetration ща the pest to the north, but not about the threat of outbreaks there.

lines 191–196. Thresholds of development are different for different stages and for populations of the same species. If the developed caterpillars hibernate in the egg, then they are ready to hatch at any moment when they get into conditions with a suitable temperature. However, in each stand, those individuals survive that hatched during the leaf development. If the population stays on this stand for a sufficiently long time, then the development of most individuals is adapted to the timing of foliage development (see commentary to lines 71–74).

Pantyukhov can be added to the references on the thresholds for L.dispar development.

Pantyukhov G. A. The influence of positive temperatures on various geographical populations of Euproctis chrysorrhoea L. and gypsy moth Lymantria dispar L. (Lepidoptera, Orgyidae). Entomological Review. 1962. 41 (2): 274–284.

Table 2. lines 316–317. The data is very interesting, especially for Ekaterinburg.

It would be necessary to explain in the “Methods” or in the “Discussion” why the SET of summer-autumn embryo development was used in this study, to describe the methodology, or to give a reference.

Line 335. Discussion. “What are the possible causes of this phenomenon?”

– which phenomenon do you mean?

Line 351. It would be necessary to explain in the “Methods” or in the “Discussion” why the SET of summer-autumn embryo development was used in this study, to describe the methodology, or to give a reference.

Conclusions. Lines 447-449. “Thus, we argue that this feature of spongy moths from northern Eurasia poses a significant invasive threat to North America in terms of enhanced risks for rapid northward range expansion”.

– It was not studied there!

Fig. 1. L.Dispar – must be L.dispar

line 456 –  Pnomarev –must be Ponimarev

References.

Latin names of insects and plants must be italics (4,5,6,9,10,11,14,22–27,30,31,34,43,44)

Some words in titles are written with capital letters (1,6,7,45,47–49), the years in the beginning, in parentheses etc. (45).

Translation of Russian titles of the papers must be corrected (11, 16 – mass propagation of ...foliage browsing insects?, leaves and needles gnawing insects?)

Author Response

Response to Reviewers’ comments

We kindly thank the referees for their detailed assessments of our manuscript and we are grateful that both found our work relevant. Please find below our responses (in italics) to the comments made by referee #2:

Referee #2

1)The research topic of the authors is relevant. The scope of work is large. However, there are some comments on the presented material. Marked in yellow in the file.

lines 37–39. «Synchronization of flight at different latitudes of the range is associated with an acceleration of larval development in northern Eurasian populations»

– that's a question! dates of summer phenomena are very close at different latitudes within a territory with a pronounced change of seasons, where the timing and rate of development of plants and insects are determined by both temperature and photoperiodic response

This is a result of our studies and result of the analysis of literature data. Concerning the spongy moth there is no evidence of strong photoperiodic effect on larval development. Such effect is noticed more often for species that overwinter as larvae.

lines 39–41. «Similar changes in developmental rate along a latitudinal gradient have not been documented for North American populations. Thus, we argue that this feature of spongy moths from northern Eurasia poses a significant invasive threat to North America in terms of enhanced risks for rapid northward range expansion».

– The range of the gypsy moth will still move north, provided there is sufficient heat and fodder plants.

Martemyanov version: Because this is an abstract, we just briefly summarize our conclusions and discussion about the findings. A more detailed explanation of this is provided in the body of the article. Yes, as a result of global warming, the spongy moth will indeed expand its range to the north. However,  our focus, here, is on population differences in the speed of the reaction to warming trends. This conclusion is based on the results of current MS, as well as on an article referenced within the MS [i.e. 24].

2) lines 71–74. “...we can say that spongy moth egg hatch typically takes place within a ~60-day time window spanning April and May, at least in the southern and central portions of the species’ range”.

Within the entire range, the timing of larvae hatching is synchronized with the host plant foliage appearance. This occurs approximately after a stable temperature transition over 10 ° C. Within the stands, these dates may differ by several days, depending on the depth of soil freezing, the timing of its thawing, the beginning of the sap flow of trees, and the structure of stands. If the moths flew into another plantation, then in the next generations those caterpillars will survive, which hatching coincided with the leaves blooming in this particular stand.

Therefore, the sum of temperatures before the flight must be calculated not from the beginning of the year, but from the date of larvae hatching, approximately after the date of the stable temperature transition through 10 ° C. In each stand, the exact timing of larvae hatching can be determined by phenological indicators, for example, birch leaves blooming, Taráxacum flowering ...

Summer events (flight of L.dispar) generally vary little at different latitudes, to a greater extent within one stand.

Martemyanov version: in this paragraph we described variation in hatching time as a function of a latitudinal gradient. Of course, the hatching of larvae, even within one stand, will vary depending on exposure to weather conditions and/or other microhabitat conditions. Does the referee mean that we should mention that hatching time depends on microhabitat conditions to a greater extend than on latitude (heat availability) in a large-scale analysis such as ours? If that’s the case, we disagree with the reviewer on this point.

3) lines 89–90. “The above data show a general trend for a shift in egg hatch and flight phenology towards later dates with increasing latitude”

– Spring in the north starts later, but the temperature rises faster than in the south, so by the middle of summer many phenomena are observed almost simultaneously at different latitudes. Closer to autumn, the photoperiodic reaction can contribute to the inhibition of the development of individual instars or stages.

L. dispar is a species with an obligatory diapause that is not governed by daylength. We added a sentence in the text to this effect (Lines 108-110).

4) lines 106–108. “Thus, spongy moth phenological data in the northernmost portions of its range are essential to assessing the risks of outbreak spread following an expansion of its range”.

Expansion of the range does not mean the ability to outbreak.

We completely agree with the reviewer –range expansion definitely does not mean the ability to outbreak in new areas, but suggests an increased risk of it. To make this statement softer, we replaced the word “risk” with “possibility”.

5) lines 109–110. Purpose ... goal ... aim?

Thanks, we keep the aim

6)  there are 3 tasks here: to find the northern border, to compare the flight dates, and to compare the SETs, and in the abstract there are 2 tasks (lines 28–32), the last two tasks are combined.

Thanks for pointing this out. We’ve now combined the last two objectives in one.

7) lines 115–116. “...above the 7°C threshold required for development to the adult stage in all three regions with corresponding data on heat availability.

Using the 7°C threshold to analyze summer events is counterintuitive. It is better to use a threshold of 10°C (see above) to determine the limit of species ranges. For example, this methodological approach was applied in predicting the distribution of Emerald ash borer to Northern Europe.

Orlova-Bienkowskaja, M.J.; Bieńkowski, A.O. Low heat availability could limit the potential spread of the Emerald ash borer to Northern Europe (Prognosis Based on Growing Degree Days per Year). Insects 2022a, 13(1), 52.

Webb, C.R.; Mona, T.; Gilligan, C.A. Predicting the potential for spread of emerald ash borer (Agrilus planipennis) in Great Britain: What can we learn from other affected areas? Plants People Planet 2021, 3, 402–413.

In the Materials and Methods section, we carefully explain that there are inter-population fluctuations in this threshold (lines 186-209 of revised version).  Our calculations are based on data gleaned from the literature on the developmental temperature threshold for the spongy moth. We also added a reference to one of our recent studies that confirms the validity of the 7°C threshold. The use of an inflated temperature threshold may lead to erroneous forecasts of range expansion due to an underestimation of available heat resource. The Emerald ash borer is a concealed insect while the spongy moth lives in the open; we feel that these two species are not ecologically comparable.

8) lines 131–132. The presence of males in a trap does not mean that there may be an outbreak in this stand. On the contrary, in high population density, male chooses female visually and prefers the natural pheromone to the synthetic one.

To verify the fact of L. dispar presence in the territory, the available data are sufficient. Trends in population dynamics can be traced from data obtained over several years, which cover all phases of the outbreak. Therefore, 2 years of records in the North American continent with a gap are sufficient only to conclude about the penetration of the pest to the north, but not about the threat of outbreaks there.

We agree with this comment. We replaced “population density” with “population status” to indicate general qualitative trends in population levels. We also referred the reader to some very high trap catches during our study (several thousand moths per trap in outbreaking areas); this means that, even if there is competition between females and pheromone lures in traps, the numbers of moths caught still provide a crude estimation of the L. dispar population status.    (L134)

9) lines 191–196. Thresholds of development are different for different stages and for populations of the same species. If the developed caterpillars hibernate in the egg, then they are ready to hatch at any moment when they get into conditions with a suitable temperature. However, in each stand, those individuals survive that hatched during the leaf development. If the population stays on this stand for a sufficiently long time, then the development of most individuals is adapted to the timing of foliage development (see commentary to lines 71–74).

We broadly agree with this reviewer, but we do bring up the issues of temperature thresholds for larval and pupal development in this paragraph. The threshold for late embryonic development is very controversial and is here discussed in light of information found in the literature. However, the late embryonic SET has a weak impact on the whole developmental rate, because it makes a small contribution to the developmental SET for the entire life cycle – about 100-110 degree-days.

10) Pantyukhov can be added to the references on the thresholds for L.dispar development.

Pantyukhov G. A. The influence of positive temperatures on various geographical populations of Euproctis chrysorrhoea L. and gypsy moth Lymantria dispar L. (Lepidoptera, Orgyidae). Entomological Review. 1962. 41 (2): 274–284.

Added to the references

11) Table 2. lines 316–317. The data is very interesting, especially for Ekaterinburg.

It would be necessary to explain in the “Methods” or in the “Discussion” why the SET of summer-autumn embryo development was used in this study, to describe the methodology, or to give a reference.

Added this moment in the Methods (L 222)

12) Line 335. Discussion. “What are the possible causes of this phenomenon?”

– which phenomenon do you mean?

We revised the text as follows: “The present study raises questions as to what mechanism is responsible for the signifi-cantly reduced developmental period, up to the imago stage, reported here for spongy moths found in the northernmost portion of the species’ Eurasian range, in comparison to values obtained for populations in the central and southern regions. ”

13) Conclusions. Lines 447-449. “Thus, we argue that this feature of spongy moths from northern Eurasia poses a significant invasive threat to North America in terms of enhanced risks for rapid northward range expansion”.

– It was not studied there!

We agree with this reviewer. We have only studied Eurasian populations, but a comparison of our results on developmental SETs with North American data gleaned from the literature suggests that northern Eurasian populations develop much faster than their North American counterparts do.

>>Fig. 1. L.Dispar – must be L.dispar

Corrected. In all Figures the Latin names turn into italic

>>line 456 –  Pnomarev –must be Ponimarev

Corrected

>>References.

>>Latin names of insects and plants must be italics (4,5,6,9,10,11,14,22–27,30,31,34,43,44)

Corrected

>>Some words in titles are written with capital letters (1,6,7,45,47–49), the years in the beginning, in parentheses etc. (45).

Corrected

>>Translation of Russian titles of the papers must be corrected (11, 16 – mass propagation of ...foliage browsing insects?, leaves and needles gnawing insects?)

11 - Gninenko, Y.I., Kavosi, M.R. Lymantria dispar (Lepidoptera, Erebidae) outbreak in north Iran. Forestry Information (2), 2016, 81-89. (In Russian)

This is an official English translation of the title: http://lhi.vniilm.ru/index.php/en/gninenko-yu-i-kavosi-m-r-lymantria-dispar-lepidoptera-erebidae-outbreak-in-north-iran

  1. Ilyinskiy, A.I. Supervision, accounting and forecast of mass propagation of needle- and leaf-eating insects in the forests of the USSR. Forest industry: Moscow, USSR, 1965.

Change the title as above.